# Bioinformatics and Genomic Analyses of the Suitability of Eight Riboswitches for Antibacterial Drug Targets

**DOI:** 10.3390/antibiotics11091177

**Published:** 2022-08-31

**Authors:** Nikolet Pavlova, Robert Penchovsky

**Affiliations:** Department of Genetics, Faculty of Biology, Sofia University “St. Kliment Ohridski”, 8 Dragan Tzankov Blvd., 1164 Sofia, Bulgaria

**Keywords:** allosteric drug targets, antibacterial drug discovery, bacterial riboswitches, bioinformatics analysis, biochemical pathways, human bacterial pathogens

## Abstract

Antibiotic resistance (AR) is an acute problem that results in prolonged and debilitating illnesses. AR mortality worldwide is growing and causes a pressing need to research novel mechanisms of action and untested target molecules. This article presents in silico analyses of eight bacterial riboswitches for their suitability for antibacterial drug targets. Most bacterial riboswitches are located in the 5′-untranslated region of messenger RNAs, act as allosteric cis-acting gene control elements, and have not been found in humans before. Sensing metabolites, the riboswitches regulate the synthesis of vital cellular metabolites in various pathogenic bacteria. The analyses performed in this article represent a complete and informative genome-wide bioinformatics analysis of the adequacy of eight riboswitches as antibacterial drug targets in different pathogenic bacteria based on four criteria. Due to the ability of the riboswitch to control biosynthetic pathways and transport proteins of essential metabolites and the presence/absence of alternative biosynthetic pathways, we classified them into four groups based on their suitability for use as antibacterial drug targets guided by our in silico analyses. We concluded that some of them are promising targets for antibacterial drug discovery, such as the PreQ1, MoCo RNA, cyclic-di-GMP I, and cyclic-di-GMP II riboswitches.

## 1. Introduction

Bacterial infections are prevented and treated by antibiotic medicines. In response to these medicaments, the bacteria adapt and antibiotic resistance (AR) occurs. AR is one of the biggest dangers to health and food security. It occurs naturally and affects people of any age, country, and gender. The accelerating force of the process is the misuse of antibiotics in humans and animals [1]. The scope of AR is global because it is rising to dangerously high levels in all parts of the world. The World Health Organization (WHO) warned in 2017 that there was a need to develop new mechanisms of AR and test new bacterial molecules as targets. The report with a growing list of infections, such as pneumonia, tuberculosis, blood poisoning, gonorrhea, and foodborne diseases, was published by the WHO (https://www.who.int/news-room/fact-sheets/detail/antibiotic-resistance, accessed on 8 August 2022). The mentioned infections are becoming harder to treat, and in some cases, are impossible to treat as the antibiotics used against them have become less effective. This results in more extended hospital stays, higher medical costs, and increased mortality.

In the United States of America (USA) and Europe, the losses surpass USD 55–70 billion and EUR 1.5 billion every year [2,3]. Increased AR prevalence among bacteria represents the most significant challenge to human health [4,5,6]. Recent WHO reports show that 700,000 people have died annually worldwide [7]. The European Union (EU) reports that 5–12% of patients acquire an infection during hospitalization. Annually, an estimated 400,000 patients are infected with a resistant bacterial strain, resulting in an average of 25,000 human deaths [8] as a result of surgical site infections [9]. As a result, there is an enormous need to discover drugs or vaccines that overcome AR.

Researchers continue to test different strategies for antibacterial drug development to discover new molecular targets and preventative diagnoses [1,10,11,12,13,14,15]. They work on bacterial mRNAs with riboswitches that are gene control elements, directly binding specific metabolites and altering gene expression [16,17]. They are highly structured cis-acting elements usually located in the 5′-untranslated region (5′-UTR) of messenger RNA (mRNA) and fold into complex three-dimensional structure receptors for their target molecules [18,19,20,21].The architecture of their structure typically consists of two functional components such as the aptamer and expression platform. The aptamer forms a binding pocket that senses metabolites and can distinguish between metabolite analogs and recognize their cognate effectors with values for the apparent dissociation constant (Kd), usually ranging from lower picomolar to micromolar concentrations [22]. The different derivatives sensed by the riboswitches are amino acids, such as glycine, glutamine, and lysine; coenzymes, including adenosylcobalamin (AdoCbl), flavin mononucleotide (FMN), S-adenosyl methionine (SAM), thiamine pyrophosphate (TPP), etc.; ions, such as Mg^2+^, Mn^2+^, and F^−^; nucleotides, including adenine, guanine, and 2′-Deoxyguanosine, etc.; signaling molecules, such as cyclic-di-adenosine monophosphate (c-di-AMP), cyclic-di-guanosine monophosphate (c-di-GMP), etc.; and other metabolites. Based on this, they are classified into 36 different classes [23,24]. Each regulates a specific metabolic pathway by sensing another kind of metabolite [25]. Thus, one riboswitch can be repeated many times in one bacterial genome and found in many different bacteria. The expression platform changes its confirmation as a result of ligand binding to the aptamer and changes the mRNA expression, usually by the termination of transcription or prevention of translation.

As an extension, the riboswitches regulate gene expression by different regulation mechanisms. Three of them are cis-regulating mechanisms, transcription termination, translational prevention, and the destabilization of mRNA, and one is the trans-acting regulation of gene expression through the riboswitch [20,23]. The riboswitch control by transcription termination is preferentially done through Rho-independent terminators, except for only one known Rho-dependent termination [26,27,28]. The trans-acting regulatory mechanism is typical for the SAM riboswitches, SreA and SreB, which can function in trans and act as noncoding RNAs [29]. The same riboswitch could sense only analogs with very similar structures. Riboswitches recognize the 3D structure of the metabolites and bind them complementary to a specific ligand-binding part of their aptamer.

The precedent-reported antibiotics and the newly established metabolites sensed by riboswitches emphasize the progress in discovering novel riboswitch-targeting compounds [30]. The first tested riboswitch-targeted analogs were DL-4-oxalysine, L-aminoethylcysteine, pyrithiamine, and roseoflavin [30]. DL-4-oxalysine and L-aminoethylcysteine are lysine analogs sensed by the Lysine riboswitch (LysC) and inhibit the growth of different Gram-positive bacteria [31]. Pyrithiamine is an analog of thiamine, which has a toxic effect on fungi and bacteria in which a pyridinium ring replaces the thiazole ring. In vivo, it is pyrophosphorylase to pyrithiamine pyrophosphate, which is sensed by the Thiamine pyrophosphate riboswitch, causing the cell to cease the synthesis and import of TPP, resulting in the death of the cell. Another thiamine pyrophosphate analog, which binds to *the thi-box* riboswitch, is the S-benzoyl thiamine monophosphate known as benfotiamine [32]. Lately, information has been published about a new method for regulating the thiamine pyrophosphate riboswitch involved in the action of nalidixic acid [33]. Nalidixic acid is an analog of benfotiamine, which binds to the active site of the thi-box riboswitch and acts as an inhibitor compound in the growth of *Escherichia coli.* Bound to the riboswitch, it inhibits the expression of reporter genes under the control of the riboswitch.

Roseoflavin is a riboflavin and flavine monophosphate analog [34]. Sensed and bonded to the flavine mononucleotide, the riboswitch regulates all genes involved in riboflavin synthesis under the riboswitch’s control [35]. It has been one of the most effective natural antibacterial compounds [35,36]. As a promising broad-spectrum antibiotic produced by *Streptomyces davaonensis* and *Streptomyces cinnabarinus*, roseoflavin production has to increase in the organisms in which it is naturally possible, such as the heterologous host’s *Bacillus subtilis* and *Corynebacterium glutamicum* [37]. The RoF and 8-dimethyl-8-aminoriboflavin uptake in the riboflavin auxotroph human pathogenic bacteria *Listeria monocytogenes have been studied* [34]. The results showed that the bacterial protein Lmo1945 is responsible for the riboflavin uptake and analogs. The computational and molecular approaches have led to the identification of an increasing number of existing and novel riboswitches [38,39,40]. The bacterial riboswitches as allosteric molecular targets can be used for the development of novel small molecule targeting, ASO targeting, and high-throughput screening (HTS) [41,42,43].

Recently, new chimeric antisense oligonucleotides targeted to glmS riboswitches have been shown to block the synthesis of glucosamine-6-phosphate to inhibit the growth of *Staphylococcus aureus* and bacterial growth [44]. Other ASOs have been engineered to target flavin mononucleotide riboswitches in *Escherichia coli*, *Listeria monocytogenes*, and *Staphylococcus aureus* [45]. The two mentioned types of research confirm that antisense oligonucleotides are suitable molecules for antibacterial drug discovery.

ASO targeting is a treatment model involving specifically designed antisense oligonucleotide (ASO) molecules [46]. They are complementary to specific targeted regions of the messenger RNA (mRNA), which means that it works on the primary structure, i.e., sequence [47]. ASO can alter mRNA expression by different previously mentioned mechanisms, including RNAse H, RNase P, and others [23]. Other ASOs have been tested on human ASOs targeting human telomerase mRNA, increasing the therapeutic effects of radiation and chemotherapy for patients with nasopharyngeal carcinoma, or c-myb ASOs, which increase the human colon cancer cell’s sensitivity to cisplatin [48,49].

The in vitro HTP for ligand binding is an automated approach based on various fluorescence detection techniques. It provides the simultaneous testing of many chemical compounds [25]. For example, fully automatable HTS to identify the compounds that bind to the glmS riboswitch has been designed to advance the development of novel antibacterial drugs targeting RNA [38]. Ribozyme-based in vitro HTP for ligand binding has been engineered to cleave external RNA molecules (Figure 1C). Applying the fluorescence resonance energy transfer method allows the detection of cleavage. The 5′-and-3′-end of nucleic acid has been attached to a reporter and a quencher. If the ribozyme is designed to cleave RNAs in the inactive form, the substrate is not cleaved and both the reporter and the quencher are detected in one molecule. In the activated state, the ribozyme detects the effector and cleaves the nucleic acid. As a result, the reporter is not quenched, and the fluorescence signal is detected [25].

Not all the found and established riboswitches are promising drug targets [8]. Therefore, using genome-wide bioinformatics analysis, we have analyzed the cyclic-di-GMP-I, cyclic-di-GMP-II, fluoride, glycine, Mg^2+^, Mn^2+^, MoCo RNA, and PreQ1 riboswitches’ distributions and functions and the biochemical pathways that are under their control in human pathogenic bacteria.

## 2. Results

One of the most significant benefits of using bacterial riboswitches is that they have not yet been found in the human genome but are widespread in human bacterial pathogens [23]. Therefore, we examined eight riboswitch classes, including c-di-GMP I, c-di-GMP II, fluoride, glycine, Mg^2+,^ Mn^2+,^ MoCo RNA, and PreQ1, to see if they are involved in the synthesis of the essential metabolites of the biochemical pathways of human bacterial pathogens and if they control an active protein transporter (Table 1).

In vitro experiments in Penchovsky’s laboratory with antisense oligonucleotides, for the first time designed with cell-penetrating peptides and targeted to GlmS [44] and FMN riboswitches [45] in three different bacteria, demonstrated that our previous genome-wide bioinformatics results [8] were proved true and could be used as a milestone for the antisense technology for antibacterial drug development.

Following the logic of our previous research for the genome-wide analyses of the riboswitches TPP, FMN, GlmS, lysine, cobalamin, SAM, SAH, adenine, and guanine, we grouped the riboswitches into four categories based on their suitability as antibacterial drug targets [8,44,45,50]. The ‘Most suitable riboswitches’ group controls the unique, essential biosynthetic pathway and the ‘Very suitable riboswitches control the essential biosynthetic pathway and protein transporter for the key metabolites. The ‘Suitable riboswitches’ are in the third category, which contains riboswitches with alternative metabolic pathways under the riboswitches’ control. These riboswitches are unsuitable and do not control synthesis but rather the degradation of metabolites (Table 2).

Furthermore, we answered the question of whether the deactivation of the function of the eight riboswitches would result in a bacteriostatic or bactericidal effect for the human bacterial pathogens or another type of bacteria based on the information that would be found. The bacterial riboswitches as allosteric molecular targets could be used for the development of novel small molecule targeting, ASO targeting, and high-throughput screening (HTS) (Figure 1) [41,42,43].

### 2.1. Most Suitable Riboswitches

#### 2.1.1. The PreQ1-I Riboswitch

The pre-queuosine-1 riboswitch class has cis-acting regulatory elements found in bacteria. The classes of the PreQ1 riboswitches are preQ_1_-I, preQ_1_-II, and preQ_1_-III. The aptamer of PreQ_1_-I is small, from 25 to 45 nucleotides long. The PreQ_1_-II riboswitch is the only one found in *Lactobacillales* with a larger aptamer with 58 nucleotides. NCBI nucleotide BLASTN analyses have proved this. The PreQ_1_-III riboswitch has an aptamer from 33 to 58 nucleotides [51]. The PreQ_1_-III riboswitch has a pseudoknot that does not incorporate its downstream expression platform at its ribosome binding site (RBS) [52]. The PreQ_1_-I is an intermediate in the queuosine biosynthetic pathway [53,54,55]. It regulates genes involved in the biosynthesis of the queuosine (Q) nucleoside from GTP due to the precise sensing and binding of preQ_1_. The PreQ1 riboswitch is found in the genomes of 15 bacteria from the 60 human pathogenic bacteria in the list of pathogenic bacteria that we have selected—*Bacillus anthracis*, *Bacillus cereus*, *Bordetella pertussis*, *Clostridium botulinum*, *Clostridium difficile*, *Clostridium perfringens*, *Enterococcus faecalis*, *Enterococcus faecium*, *Haemophilus influenza*, *Listeria monocytogenes*, *Neisseria gonorrhoeae*, *Neisseria meningitides*, *Staphylococcus epidermidis*, *Staphylococcus saprophyticus* and *Streptococcus agalactiae* (Table 1, List 1) [56]. Fasta files have been created with the sequences from all 15 species and used for detailed multiple alignments with Clustal X (Appendix A). All collected sequences have been analyzed with the Nucleotide Basic Local Alignment Tool from the National Center for Biotechnology Information (NCBI) (https://blast.ncbi.nlm.nih.gov/Blast.cgi accessed on 15 June 2022). The analysis showed that the PrerQ1-I riboswitch of each human pathogenic bacteria is not found in the human genome nor the genome of probiotic bacteria from the intestinal tract (Appendix A). Some of these are available in the RSwitch database (https://penchovsky.atwebpages.com/results_database3.php accessed on 15 June 2022).

The first reports for the preQ1 riboswitch were from 2004 when it was located in the ykvJKLM (queCDEF) operon of the *Bacillus subtilis*, which encodes four genes for queuosine production [53,57]. When PreQ_1_ is bound to the riboswitch aptamer domain, it results in transcription termination within the leader to downregulate the expression of the four genes. After that, the preQ_1_ riboswitch was found as a conserved sequence on the 5′-UTR of genes in many Gram-positive bacteria, regulating the synthesis of preQ_1_ [58]. It regulates gene expression by preventing the transcription and termination of the translation. PreQ_1_-mediated transcriptional attenuation is controlled by the anti-terminator and terminator hairpin in the riboswitch [59]. For example, in *Bacillus subtilis*, the anti-terminator is less stable than the terminator [59]. In the presence of preQ_1_, the 3′-end of the adenine-rich domain interacts with the center of the P1 hairpin loop and forms an H-type pseudoknot [59]. In the native mRNA structure, the specific binding of preQ_1_ forms a terminator hairpin, causing the RNA polymerase to prevent transcription. This is known as the OFF regulation of gene expression or transcription [60]. The second type of regulation is the termination of translation. The protein translation in prokaryotes is initiated by binding the 30S ribosomal subunit to the Shine–Dalgarno (SD) sequence in mRNA. The termination of transcription by the riboswitch is a result of blocking the Shine–Dalgarno sequence of mRNA to prevent the binding of the ribosome to mRNA for translation. When the aptamer domain senses and binds the PreQ1, this promotes the sequestration of a part of the SD sequence at the 5′-end of the P2 stem. This causes the inaccessibility of the SD sequence due to the formation of the pseudoknot in the presence of preQ_1_ and shows the OFF regulation of genetic expression in the translational riboswitch or inhibition of translational initiation [61,62]. In eubacteria, de novo queuosine biosynthesis requires synthesizing the free nucleobase from GTP (Figure 2) [53,63].

The first known intermediate in the biosynthetic pathway is the 7-cyano-7-deazaguanine, also known as PreQ0, converted to preQ1 in an NADPH-dependent reduction catalyzed by QueF [64]. In a guanine exchange reaction, the preQ1 is inserted at the appropriate position in the anticodons of the relevant tRNAs. After that, queuosine production continues in situ, first with the addition of an epoxycyclopentandiol ring derived from the ribosyl moiety of S-adenosylmethionine [65]. The *queC*, *queD*, *queE*, and *queF* gene families are commonly associated with queC motifs in biochemical steps upstream of preQ1 and are under the control of the riboswitch. A group of small synthetic molecules specifically and reversibly bind to PreQ1 with high affinity inducing conformational change and modulating the riboswitch’s activity through transcription termination [66]. Thus, structure- and target-based approaches can be used to identify the mechanism of synthetic ligands that bind to and regulate the complex, folded RNAs [41]. We conducted a detailed structural analysis of the PreQ1-I riboswitches found in *Bacillus anthracis*, *Bacillus*
*cereus*, *Bordetella pertussis*, *Clostridium botulinum*, *Clostridium difficile*, *Clostridium perfringens*, *Enterococcus faecalis*, *Enterococcus faecium*, *Haemophilus influenza*, *Listeria monocytogenes*, *Neisseria gonorrhoeae*, *Neisseria meningitides*, *Staphylococcus epidermidis*, *Staphylococcus saprophyticus*, and *Streptococcus agalactiae.*

The multiple alignments and nucleotide blast analyses of the sequences of the riboswitch from the different bacteria and the human genome showed that between the bacterial sequences are specific regions that are not found in the human genome or probiotic bacteria and could be used as a potential target of the riboswitch-based bacterial drug compound. The results obtained from bioinformatics and the genomic analysis of its sequences in bacteria can be used to determine one or more specific motifs that will serve as a target for new antibacterial agents. These agents may be antisense oligonucleotides (ASOs) with a complementary target sequence to the target and a peptide to ensure their penetration into the bacterial cell (pVEC or other cell-penetrating protein). After specific binding of the target sequence to the ASO, the gene expression of the gene under the riboswitch’s control will be regulated. The PreQ1 riboswitch is classified as the most suitable riboswitch because it regulates the gene expression of the essential and transporter proteins for the essential metabolites and alternative biosynthetic pathways (Table 2). If the vital gene expression is blocked, the synthesis of the guanosine nucleoside cannot occur, resulting in its deficiency.

#### 2.1.2. The MoCo RNA Motif Riboswitch

The molybdenum cofactor motif (MoCo RNA motif) riboswitch is a cis-acting regulatory element with a conserved RNA structure that senses and binds specifically to the molybdenum cofactor or the tungsten cofactor [67]. The molybdenum cofactor (MoCo) is a tricyclic pyranopterin coordinating an atom of molybdenum (Mo)–transition metal. Analyzing the biosynthesis, we see that MoCo is inserted into the active site of the Mo-dependent enzymes that control the redox activity of Mo to catalyze critical reactions in the carbon, nitrogen, and sulfur metabolic cycles [68]. Based on previously published genetics experiments, the motif is a regulatory element responsible for charging concentrations of molybdenum and tungsten cofactors [68]. The biosynthesis of the molybdenum cofactor in bacteria is fully known. The molybdenum cofactor (MoCo) could be inserted directly into the specific molybdoenzymes and modified by adding nucleotides to the molybdopterin phosphate group [69]. Structural studies of MoCo associated with liver sulfite oxidase revealed a unique pterin component termed molybdopterin (MPT) [70,71]. After that, different MPT-dinucleotide variants of MoCo were identified in bacterial molybdoenzymes, including the molybdopterin guanine dinucleotide (MGD) form found in *Escherichia coli*. The cyclic pyranopterin monophosphate (cPMP) is a biosynthetic precursor of the dithiolene group bearing molybdopterin. The biosynthesis of MoCo is an ancient, ubiquitous, and highly conserved biosynthetic pathway leading to the activation of molybdenum. It is the essential component of a group of redox enzymes causing transformations. The biosynthesis of Moco in bacteria can be separated into four reactions. The first one is the formation of the cyclic pyranopterin monophosphate.

The pyranopterin ring is constructed during MoCo biosynthesis by a complex rearrangement of guanosine 5′-triphosphate (GTP) into cyclic pyranopterin (cPMP). The reaction is mediated by two enzymes, MoaA and MoaC, that is, molybdenum cofactor biosynthesis protein A (MoaA) and molybdenum cofactor biosynthesis protein C (MoaC). MoaA catalyzes the transformation with fine accompaniment or without MoaC (Figure 3) [72]. The second reaction is the formation of MPT. We observed the third reaction, which is the insertion of molybdenum into molybdopterin to form MoCo. In the end, we observed the modification of MoCo with the attachment of GMP or CMP to the phosphate group of MPT. The first three reactions were under the control of the MoCo RNA riboswitch. The last reaction caused the formation of the dinucleotide variant of MoCo [73]. More than 50 molybdoenzymes have been identified in bacteria to date.

The MoCo RNA riboswitch is found in 10 pathogens from the list of 60 human bacterial pathogens presented in the Appendix A. It is located in the genomes of *Clostridium perfringens*, *Salmonella enterica*, *Salmonella typhi*, *Shigella sonnei*, *Enterobacter* sp., *Escherichia coli*, *Haemophilus influenza*, *Klebsiella pneumonia*, *Vibrio Cholerae*, and *Yersinia pestis.* Five are on the list of bacteria published by the WHO as being in urgent need of new antibiotics. Based on the multiple alignments, we found specific regions that were not found in the human genome after the BLAST search and can be used as a potential target of the riboswitch-based bacterial drug compound. The results are available on the RSwitch database (https://penchovsky.atwebpages.com/results_database4.php, accessed on 17 August 2022).

Based on the results, MoCo RNA is one of the most suitable riboswitches because it controls the biosynthesis of essential metabolites. There are no transporter proteins that are not under the riboswitch’s control or alternative biosynthesis (Table 2). If the genes responsible for the essential metabolites are targeted with ASOs, their gene expression will be terminated and the metabolites will not be synthesized. If one of the three reactions under the control of the MoCo RNA riboswitch is blocked, folate synthesis will not be possible.

### 2.2. Very Suitable Riboswitches:

#### Cyclic-Di-GMP-I and Cyclic-Di-GMP-II Riboswitches

Cyclic di-guanylate di GMP (cyclic di-GMP or c-di-GMP) is a second messenger used in signal transduction in bacteria that links changes in environmental cues with the regulation of umpteen phenotypes including and not limited to biofilm formation, DNA repair, virulence, and motility [74,75]. Various c-di-GMP synthesis and degradation enzymes are present in different bacteria. Each of the enzymes responds to an additional signal integrated into the changes in the levels of cyclic di-GMP. It regulates the downstream phenotypes by a variety of different mechanisms. One way is the control of the initiation of transcription via direct interaction with transcription factors. The second way is to bind to RNA riboswitches and control gene expression post-transcriptionally. The third way is directly integrating enzymes or proteins to regulate their activity allosterically [74].

The first class of c-di-GMP-sensing riboswitches is the cyclic-di-GMP-I riboswitches. They sense and bind cyclic di-GMP, which is used in many microbial processes including biofilm formation, virulence, and motility [75]. This class was identified as a conserved RNA-like structure (“GEMM motif”) by bioinformatics [67]. Cyclic di-GMP-I riboswitches are present in many bacteria and are most common in clostridia and certain varieties of proteobacteria. They are present in the genomes of seven different types of human pathogenic bacteria (Table 1)—*Bacillus anthracis*, *Bacillus cereus*, *Clostridium botulinum*, *Clostridium difficile*, *Clostridium perfringens*, *Clostridium tetani*, and *Vibrio cholerae.* We have conducted detailed, sequence-based analyses with the downloaded sequences from the Rfam database as with the previous two riboswitches. The cyclic di-GMP-I riboswitch is the first to be discovered as part of signaling and it does not have a primary role in regulating metabolism. The three-dimensional structures of cyclic di-GMP-I riboswitches have been determined using X-ray crystallography [76,77]. Some of the c-di-GMP-I riboswitch homologs recognize the molecule of cyclic AMP-GMP [78,79].

The second class of c-di-GMP sensing and binding riboswitches is the cyclic-di-GMP-II riboswitches (c-di-GMP-II riboswitches). They are not related to cyclic-di-GMP-I riboswitches structurally but their function is equal. Similar to the previous class, they were discovered by bioinformatics tools and are typical for bacteria such as clostridia and deinococcus. The c-di-GMP-II riboswitches are found in the genomes of four bacteria infecting humans—*Clostridium botulinum*, *Clostridium difficile*, *Clostridium perfringens*, and *Clostridium tetani* (Table 1). We observed that both riboswitch classes—the c-di-GMP-I and the c-di-GMP-II—were common in clostridia. In *Clostridium difficile* strains, the cyclic di-GMP-II riboswitch is found adjoining a group-I catalytic intron. Their group includes ribozymes that catalyze the splicing of the RNA molecule. In the riboswitch-associated case, the outcome of the splicing reaction catalyzed by the intron is controlled by the riboswitch in response to cyclic di-GMP levels [80]. Cyclic di-GMP-II riboswitches have pseudoknotted structures, and several nucleotide positions are highly conserved.

The c-di-GMP is an intracellular signaling molecule that regulates many processes in bacteria, with a central role in controlling the switch between motile and nonmotile lifestyles. Because c-di-GMP affects motility and adherence, it can influence bacterial pathogenicity [81,82]. For example, *Clostridium difficile* c-di-GMP regulates swimming and surface motility, biofilm formation, toxin production, intestinal colonization, and cell envelope proteins [83]. In 2018, the functionality of 11 c-di-GMP riboswitches was confirmed, regulating independent downstream gene expression of the upstream promoters. The research showed that the first class of c-di-GMP riboswitches uniformly functions as “off” switches in response to c-di-GMP, whereas the second class of c-di-GMP riboswitches acts as “on” switches [83]. Transcriptional analyses of genes of c-di-GMP riboswitches over a broad range of c-di-GMP levels showed relatively modest changes in c-di-GMP levels that can alter gene transcription concomitant effects on microbial behavior. The bacteria have developed a specific signal transduction system involving multiple diguanylate cyclases (GGDEF) and phosphodiesterase domain-containing proteins (EAL/HD-GYP) (Figure 4).

They modulate the levels of the same diffusible molecule of c-di-GMP to transmit signals and obtain specific cellular responses [84,85]. Understanding signal transduction and the biochemical connection between a sensor and the proteins it regulates is crucial. As a result of the response to extracellular stimuli, bacteria alter intracellular c-di-GMP concentrations through the opposing activities of diguanylate cyclases (DGCs) that synthesize c-di-GMP from guanosine triphosphate (GTP) and phosphodiesterases (PDEs) that hydrolyze c-di-GMP [81]. Most enzymes are predicted to be membrane-localized, and several contain additional domains that may modulate enzymatic activity [86]. If and how these features impact c-di-GMP synthesis and hydrolysis is challenging to predict and must be determined experimentally.

Recently, biosensors based on the cyclic di-GMP riboswitches have been used for the real-time imaging of in vivo enzyme activity related to cyclic di-GMP signaling [87,88]. Since the fluorescence activation mechanism is not dependent on the host’s gene expression machinery, the biosensor should be more portable between different bacteria.

The cyclic-di-GMP-I and cyclic-di-GMP-II riboswitches are very suitable because they control both the synthesis and the degradation of essential metabolites for the bacteria (Table 2). There are published results that show that in Gram-negative bacteria, the production of the signal molecule c-di-GMP by diguanylate cyclases (DGCs) is the essential stimulus for biofilm formation, causing the development of chronic infections. The DGCs have been tested as targets for new chemotherapeutic drugs with anti-biofilm effects. It was found that azathioprine—an immunosuppressive drug used to treat Crohn’s disease—inhibited WspR-dependent c-di-GMP biosynthesis in bacterial cells [89]. The medicament contained an inhibitor of 5-aminoimidazole-4-carboxamide ribotide transformylase (AICAR), an enzyme involved in purine biosynthesis. The researchers suggested that inhibition of c-di-GMP biosynthesis by azathioprine may be due to the perturbation of intracellular nucleotide pools [89]. The WspR is a conserved GGDEF domain-containing response regulator in Gram-negative bacteria, bound to c-di-GMP at an inhibitory site [90]. The activity of WspR is abolished in an E. coli pure mutant strain, unable to produce AICAR transformylase. As a result, the azathioprine did not prevent biofilm formation by *Pseudomonas aeruginosa* [89]. Our results indicate that azathioprine can prevent biofilm formation in *E. coli* by inhibiting c-di-GMP biosynthesis, suggesting that such inhibition might contribute to its anti-inflammatory activity in Crohn’s disease.

The multiple alignments and BLAST analyses showed that none of the bacterial c-di-GMP was found in the human genome and the genome of the *Lactobacillus* sp. Visualization of the results is presented on RSwitch at https://penchovsky.atwebpages.com/results_database3.php (accessed on 15 June 2022). These results confirm the suitability of the riboswitch as a suitable target for new drug development.

### 2.3. Suitable Riboswitches

#### Mg^2+^ Sensor Riboswitch

The magnesium sensor (Mg^2+^ sensor) riboswitch is a cis-regulatory element located in the 5′-UTR of the *mgtA* magnesium transporter gene. The riboswitch regulates the expression of the protein MgtA, which is a magnesium transporter. The Mg^2+^ is the most plenteous bivalent cation in biological systems. It is obligatory for ATP-mediated enzymatic reactions, ribosomes, and membrane stabilization [91,92]. The Mg^2+^ riboswitch is found in the genomes of six different types of bacteria—*Enterobacter* sp., *Escherichia coli*, *Klebsiella pneumoniae*, *Salmonella enterica*, *Salmonella typhi*, and *Shigella sonnei* (Table 1). Four of the mentioned bacteria are on the list of WHO bacteria for which there is an urgent need for new antibiotics. The *Salmonella enterica* serovar typhimurium, which is an enteric bacteria, possesses three different Mg^2+^ transporters and a regulatory system—PhoP/PhoQ, whose activity is regulated by the extracytoplasmic levels of Mg^2+^ (Figure 5) [92,93]. One of the PhoP/PhoQ regulatory system responses includes the induction of the transcription of the *mgtA* and *mgtCB* genes, which encode two magnesium transporters [94]. The 5′-UTR of the *mgtA* gene adopts different stem-loop structures depending on the levels of magnesium, and as a result, this is the first example of a cation-sensing riboswitch. The initiation of *mgtA* transcription responds to extracellular Mg^2+^ levels and elongation into the coding region to cytoplasmic Mg^2+^. One ligand—the Mg^2+^—is sensed by different cellular compartments to regulate disparate steps in gene transcription. The PhoP-activated Mg^2+^ transporter mgtB is also regulated by Mg^2+^ [91,92]. The results were previously presented that show that the Mg^2+^ riboswitch targets the mgtA transcript for degradation by RNase E when the cells are grown in a rich medium of Mg^2+^ [94].

The presence of the 5′-UTR modulates the expression of *mgtA* and *mgtCB; the* MgtB regulation is more complex and in low Mg^2+^, the induced expression of *mgtCB* leads to the expression of the downstream protein MgtB but not of MgtC. The riboswitch is in the group of suitable riboswitches for antibacterial drug discovery. It controls the transport of essential ligands for the bacteria (the magnesium bivalent cation) and regulates the genes involved in the biosynthesis of other essential metabolites (Table 2). It could also be in the previous category with the very suitable riboswitches but the bacteria transport of the Mg^2+^ via other protein transporters is controlled by other riboswitches such as M-box. In this case, only one ASO is not enough to affect bacterial growth and life. If only the synthesis is blocked, the transport could compensate for the deficiency in essential metabolites.

The multiple alignments are available at https://penchovsky.atwebpages.com/results_database2.php (accessed on 19 August 2022). Similar to these results, bacterial riboswitches were not found in the human genome, which means that the first important rule for their possible application has been observed.

#### 2.4. Not Suitable Riboswitches

##### 2.4.1. The Glycine Riboswitch

The glycine riboswitch is an RNA regulatory element that senses and binds glycine. Glycine is one of the amino acids. The riboswitch usually consists of two tandem metabolite-binding aptamer domains and an expression platform [95,96,97]. The complex architecture of most members of the glycine riboswitch class interfered with initial attempts to prove that glycine was being directly sensed [98]. The aptamers bind glycine to regulate the expression of downstream genes. In *Bacillus subtilis*, this riboswitch is found upstream of the *gcvT* operon and regulates glycine degradation. When glycine is in excess, it binds to both aptamers, activates these genes, and forces glycine degradation [26]. In vivo experiments have already demonstrated that glycine does not need to bind both aptamers for regulation [99,100]. A mutation in the first aptamer causes a more significant reduction in downstream gene expression, whereas mutation in the second has varying effects. Bioinformatics analyses of tandem glycine riboswitches revealed that the two binding pockets are differentially conserved between ON (transcription termination) and OFF (prevention of translation) switches [97]. The ON switch variants prefer to bind to the first aptamer and promote helical switching, whereas the OFF switch variants bind to the second aptamer [97]. Scientists have observed the cooperative effect between the two aptamers on the glycine response when both binding pockets are maximally stabilized with three GC base pairs [97].

The glycine-regulated expression of the *gcvT* operon is needed for the growth, swarming motility, and biofilm formation in *Bacillus subtilis* [101]. In Bacillus subtilis, the three-gene operon gcvT-gcvPA-gcvPB catalyzes the initial reactions to use glycine as an energy source. Two forms of the gcvT RNA motif, types I and II, have been identified. They are represented by the region’s architecture immediately upstream of the VC1422 gene, which encodes putative sodium and the alanine symporter. The gcvT is an aminomethyltransferase called glycine cleavage system protein T and catalyzes the reaction: (6S)-5,6,7,8-tetrahydrofolate + (R)-N^6^-(S^8^-aminomethyl-dihydro lipoyl)-L-lysyl-[protein] --> (6R)-5,10-methylene-5,6,7,8-tetrahydrofolate + (R)-N^6^-dihydrolipoyl-L-lysyl-[protein] + NH_4_^+^ according to UniProt. The glycine cleavage system is a catabolic mechanism for glycine. There are three key glycine catabolism enzymes, that is, glycine decarboxylase (GLDC), glycine cleavage system H protein (GCSH), and glycine N-acyltransferase (GLYAT). GLDC is a key component of the highly conserved glycine cleavage system in amino acid metabolism, which catalyzes the breakdown of glycine to form CO_2_, NH_3_, and 5,10-MTHF (Figure 6). 

The glycine riboswitch is found in 24 different human bacterial pathogens. It is found in *Acinetobacter baumannii*, *Bacillus anthracis*, *Bacillus cereus*, *Bordetella pertussis*, *Brucella abortus*, *Clostridium botulinum*, *Clostridium difficile*, *Clostridium perfringens*, *Clostridium tetani*, *Haemophilus influenzae*, *Listeria monocytogenes*, *Mycobacterium tuberculosis*, *Mycobacterium ulcerans*, *Neisseria gonorrhoeae*, *Neisseria meningitidis*, *Pseudomonas aeruginosa*, *Staphylococcus aureus*, *Staphylococcus epidermidis*, *Staphylococcus saprophyticus*, *Streptococcus agalactiae*, *Streptococcus mutans*, *Streptococcus prenumoniae*, *Streptococcus pyogenes*, and *Vibrio cholerae*. Two of them are on the WHO’s list for priority antibiotic discovery.

The glycine riboswitch is one of the riboswitches for which the study and description are prolonged and difficult because of the complex character of the action [102]. Currently, we classify it as an unsuitable riboswitch based on the previously mentioned facts and because it regulates the degradation of essential metabolites for the bacteria rather than the synthesis (Table 2).

##### 2.4.2. The Mn^2+^ Riboswitch

The manganese bivalent cation riboswitch (Mn^2+^ riboswitch, YybP-YkoY leader RNA element) is a regulatory element encoding the manganese (Mn) efflux pump in different bacteria [103]. We found it in the genomes of 14 bacteria—*Acinetobacter baumannii*, *Bacillus anthracis*, *Bacillus cereus*, *Bordetella pertussis*, *Enterobacter* sp., *Escherichia coli*, *Listeria monocytogenes*, *Pseudomonas aeruginosa*, *Salmonella enterica*, *Staphylococcus aureus*, *Staphylococcus epidermidis*, *Streptococcus pneumoniae*, *Vibrio cholerae*, *and Yersinia pestis* (Table 1). Four bacteria are on the priority list published by the WHO for discovering new antibacterial agents. The yybP-ykoY riboswitch is formed by a four-helix (P1, P2, P3, and P4) aptamer with a transition metal-sensing pocket formed by nucleotides in loops in P1 and P3 [104] and the expression platform. The yybP-ykoY motif is upstream of genes that encode uncharacterized or unsatisfactorily characterized proteins related to Mn^2+^ homeostasis. The MntP of Escherichia coli and YaoB of *Lactococcus lactis*, which are yybP-ykoY-regulated gene products, have been independently implicated as Mn^2+^ efflux transporters capable of alleviating cellular Mn toxicity [104,105]. There are associated yybP-ykoY family riboswitches that respond to higher Mn^2+^ levels in vitro and in vivo by modulating and translating downstream genes or genes within the transcript [104,106,107]. The manganese (Mn) homeostasis support is essential for different bacterias’ virulence. For example, the PsaBCA Mn importer, the MntE exporter, and the PsaR transcription regulator establish the manganese homeostasis of *Streptococcus pneumoniae*. The MntR protein detects the level of Mn in the cell and acts as a transcription factor to control the expression of MntH, MntP, and MntABCDE, which are manganese transporters. The regulation of *mntR* binds to the mntP promoter (Figure 7a). The mgtA (aptamer domain) adopts a canonical yybP-ykoY structure with a three-way junction that is formed in the presence of bivalent calcium cations (Ca^2+^) or manganese bivalent cations (Mn^2+^) at a physiological Mg^2+^ concentration [103]. The Ca^2+^ binds to the RNA aptamer with a higher affinity than Mn^2+^. In vitro, the activation of the transcription of mgtA by Mn^2+^ is more prominent than by Ca^2+^. As a result, the yybP-ykoY riboswitch in *Staphylococcus pneumoniae* regulates the Ca^2+^ efflux [103].

When extracellular concentrations are low, bacteria develop efficient manganese procurement systems under transcriptional control to assume manganese. Some bacteria possess manganese-specific export systems to neutralize it when it is overloaded. They are essential for managing intracellular manganese levels in vivo [108]. High appetency Mn^2+^ uptake by MntH is substantially greater than the uptake by MgtE. There is a suggestion that MgtE contributes lightly to Mn^2+^ uptake in wild-type cells [109]. Low magnesium media gates wild-type MgtE as the MgtE suppressor. This leads scientists to believe that the MgtE channel can allow entry to Mn^2+^ even when Mg^2+^ is in excess. The restoration of part of the Mn^2+^ concentration and pyruvate kinase and superoxide dismutase effect in the *mntH* strain against Mn distress did not require *mgtE*. As a result, Mn^2+^ entered the cells by a mechanism in addition to MgtE transporting Mg^2+^ [109]. In Hohle and O’Brian’s research, the growth defect created by the magnesium deficiency was partially recovered by lowering manganese in the growth medium, suggesting that manganese can be toxic under low magnesium conditions. They found that manipulating Mn^2+^ and Mg^2+^ levels in the medium could affect growth without changing the intracellular concentrations of the metals in the bacteria [109].

The manganese riboswitch regulates the gene expression of manganese transporters. They are essential for the virulence and ability of the bacteria to react to oxidative stress and manganese toxicity (Table 2). Based on this, we classified this riboswitch into the unsuitable riboswitches group. It could be moved to the suitable riboswitches group if we focus on the toxicity of the lower levels of manganese.

##### 2.4.3. The Fluoride Riboswitch

The fluoride riboswitch is an RNA motif with a conserved RNA structure that senses and specifically binds fluoride ions [67,110,111]. Fluoride is the 13th most abundant element [110]. This riboswitch class increases the expression of downstream genes when fluoride levels are raised. Thus, the genes reduce the toxic effect of fluoride at high levels. Many genes are under the control of the fluoride riboswitches. Two are the *CrcB* and *EriC (CIC)* genes, which encode proteins that remove fluoride from the bacterial cell. ClC proteins have been shown to function as fluoride-specific fluoride/proton antiporters [110,111]. The structure of the riboswitch is already known [112]. The guiding research of the thiamine pyrophosphate riboswitch (TPP) structure helped to characterize the fluoride riboswitch’s interactions with fluoride as well as its structure. The fluoride riboswitch has two helical stems adjoined by a helical loop with the capacity to become a pseudoknot [112]. The fluoride ligand is bound within the center of the riboswitch fold and is surrounded by three Mg^2+^ ions. The Mg^2+^ ions convert the fluoride ions into the negatively charged *crcB* RNA scaffold [112]. The fluoride riboswitches are found in many organisms including bacteria and archaea. We found them in the genomes of *Acinetobacter baumannii*, *Bacillus anthracis*, *Bacillus cereus*, *Clostridium botulinum*, *Clostridium difficile*, *Clostridium perfringens*, *Enterococcus faecalis*, *Enterococcus faecium*, *Escherichia coli*, *Klebsiella pneumoniae*, *Pseudomonas aeruginosa*, *Streptococcus mutans*
*(**S. mutans**)*, and *Yersinia pestis* (List 1, Table 1). Four of the thirteen types of bacteria are on the WHO’s priority list for urgent antibiotic development. One of the most studied is the fluoride riboswitch in *S. mutans*. It is the cause of dental caries. Sodium fluoride has inhibited the growth rate of *S. mutans* using glucose as an energy and carbon source [113]. The fluoride riboswitch clarifies the bacterial defense mechanism in acting against the toxicity of high concentration levels of fluoride by regulating the downstream genes of the riboswitch upon binding the fluoride ligand. The crcB motif RNAs were reported in 2010 for the first time and were predicted to control the expression of genes encoding K^+^ transporters, Cl^−^ transporters, and DNA repair proteins [114]. After the identification of the c-di-GMP riboswitch class, synthetic dinucleotides for binding the crcB aptamer for in-line probing in in vitro assays were designed [98,110,115].

The dinucleotide compounds did not affect the RNA. The fluoride-containing compounds were tested with potassium fluoride and sodium fluoride. The fluoride was the cognate ligand for the crcB orphan riboswitch [98,110]. The *eriC* gene encodes a homologous protein to Cl^−^ transporters [116,117]. The EriC transporters under the control of fluoride riboswitches have differences in their amino acids, determining the specific F^−^ ion export [111] (Figure 7b). The transporters EriCF and Fluc cause fluoride resistance [118]. EriCF is an F−/H+ antiporter. The fluoride exporters are significant for mediating toxic effects, causing a 200-fold increase in the resistance to bacterial growth arrest [74,98]. Both transporters–crcB and EriC—are under the gene control of the riboswitch. The genes do not encode the essential metabolites for the bacteria, but they are crucial for overcoming fluoride toxicity. The riboswitch is classified as unsuitable (Table 2). Despite being in the inappropriate category, we suggest that it not be excluded entirely from forthcoming studies because the export and import of fluoride regulated by the riboswitch in large F^−^ concentrations leads to the toxicity of bacteria.

## 3. Discussion

The ineffectiveness and the untimely administration of medications due to AR leads to a decline in the overall state of human health and even death [8]. The growing threat of AR could be regulated by cutting antibiotics and medicaments to a minimum and increasing antibacterial vaccination programs. In addition, developing new antibiotic classes dealing with multidrug-resistant pathogenic bacteria is essential [119]. All listed strategies diminish the risk of unknown drug-resistant bacteria. The riboswitches are new targets in the bacterial cells for antibacterial drugs. They are molecular sensors for binding small molecules in the cell [120]. The pathogenic bacteria have not been exposed to synthetic riboswitch-binding drug agents. The riboswitches are widespread in bacteria and have not yet been found in the human genome. For example, the cyclic-di-GMP I, cyclic-di-GMP I II, fluoride, glycine, Mg^2+^, Mn^2+^, MoCo RNA, and PreQ1 riboswitches are found in 33 pathogenic bacteria out of a list of 60 known bacterial pathogens infecting humans, as seen in the Appendix A. Some of them are found in more than one species of bacteria. Glycine is found in 24 species, PreQ-1 in 15, Mn^2+^ in 14, c-di-GMP I in 7, MoCo RNA in 10, Mg^2+^ sensor in 6, and c-di-GMP II in 4. Also, one bacterial genome could find more than one riboswitch. Their structure, conservative aptamer parts, and gene expression regulation mechanisms are well-known. They can regulate the gene expression of key metabolites and subsequent synthesis of key metabolites. They can be used as a target for new antibiotic-like molecules. The PreQ1 riboswitch is involved in the queusine nucleoside biosynthetic pathway without a transporter protein for the essential metabolites and alternative biosynthetic pathways. The MoCo RNA motif is involved in the molybdenum cofactor biosynthetic pathways without a transport protein and alternative biosynthetic pathways. The cyclic-di-GMP-I and cyclic-di-GMP-II riboswitches are genes under riboswitch regulation related to biofilm formation, virulence, and motility. The Mg^2+^ riboswitch is responsible for expressing the protein MgtA, which is a magnesium transporter. The glycine riboswitch regulates glycine degradation. The Mn^2+^ riboswitch regulates genes and encodes the manganese efflux pump in different bacteria. The fluoride riboswitch regulates the *CrcB* and *EriC (CIC)* proteins, reducing the toxic fluoride levels. Our study shows that the riboswitches that are the most suitable targets for antibacterial drug discovery are the PreQ1-I and MoCo RNA motif riboswitches. They are followed by the cyclic-di-GMP-I and cyclic-di-GMP-II riboswitches, which are very suitable. A suitable riboswitch is the Mg^2+^ riboswitch, which could be very suitable in some cases. Without synthesizing and transporting their essential metabolites or ions into the cell, the bacteria cannot synthesize proteins and regulate metabolic processes. They cannot carry out oxidation processes as well as ion toxicity. This causes their growth to be blocked, multiplied, and even perish in some cases. We encourage further laboratory analyses and investigation on the topic, using specific ASOs that hybridize with chosen target RNAs with the potential bacteriostatic or bactericidal effect against human pathogenic bacteria. None of the mentioned eight potential riboswitch-based targets have been included in an officially published clinical trial, patent, or database. We aim to select such sequences from the genome of one or more pathogenic bacteria in the region of a particular riboswitch that is not present in the genome of either a human or probiotic bacterium. There are riboswitch classes found in a restricted number of bacteria. They are suitable for designing a specific antibiotic for each of them, which will selectively target the riboswitches and not be toxic or harmful to the probiotic bacteria. We have observed significant similarities among species from the same genus based on the multiple detailed alignments. Still, in some cases, one specific riboswitch is found in only one species. When designing a specific preparation based on riboswitches, it is necessary to analyze whether this riboswitch is found in all members of the genus or only in a few species. The precisely chosen region of 16–25 nucleotides may be strictly specific for only one species and may not be effective against others of the same genus. This specific targeting would allow the administration of the ASO drug without affecting the normal human microbiome, particularly its probiotic bacteria. The antibiotics with narrow-spectrum effects target a specific sequence from the riboswitch of a bacteria and inhibit it. The antibiotics with broad-spectrum effects use the highly conserved parts of the riboswitches found in more than one pathogenic bacteria and target the riboswitches in all pathogenic bacteria in which they are detected. The newly developed antibiotics may have a bacteriostatic or a bactericidal effect on bacteria. Bacteriostatic antibiotics are less likely to cause resistance. The bactericidal antibiotics have strong selective pressure resulting from the hurried development of resistance. To survive, the bacteria must barely proceed to mutagenesis, which could be rapidly detected and overcome by redesigning the ASO. Not all riboswitches control all the biochemical pathways for synthesizing their effectors. There are alternative metabolic pathways and transporter proteins that are not controlled by the riboswitches. As a result, the bacteria will produce the essential metabolite or the transporter protein using non-riboswitch-controlled pathways. The potential risks of the side effects of ASO-based medications depend on their sequence and type of modification. Off-target effects leading to physiological changes may occur due to the non-specific binding of the drug to partially complementary sequences or an immune response may be triggered. Non-specific binding may affect the gene expression of extra-target products with side effects. To avoid these side effects, it is necessary to conduct precise analyses at the bioinformatics level if the specific target selected by the bacterial riboswitch is not detected in the human genome or, if we want to be even more precise, is not detected in the genome of probiotic bacteria. Furthermore, modifications to the second generation of ASOs such as 2′-O-methoxyethyl and 2′-O-methyl are known to reduce the toxicity of non-specific binding. The reduction in the side effects is also achieved through the correct drug method. Moreover, our laboratory has experimentally proven four of the most promising riboswitches as targets for antibacterial drug discovery, as described in the patent application [121]. Two of them are the GlmS and FMN riboswitches, which have been targeted with an antisense oligonucleotide. The results of the experimental work have been published [44,45].

There is a possibility of toxicity and cross-reactivity for drugs that target riboswitches. The riboswitch-targeting agent resistance may occur due to aptamer mutations. The point mutation should be in many copies of one certain riboswitch, which is not likely to happen. It is also possible to perform an analysis based on data from the Comprehensive Antibiotic Resistance Database (https://card.mcmaster.ca/ accessed on 15 June 2022) to check for resistant genes and related phenotypic manifestations for the presence and prediction of resistant genes and the related phenotypic manifestations of a particular bacterial strain. During the bioinformatics analysis, the entire genome of the bacterial strain will be compared for the possible presence of AR with the available database. The subsequent investigation and studies will be reasonable, using specific ASO hybridizing with target RNAs.

Suitable for penetrating agents in prokaryotic cells are cell-penetrating peptides such as pVEC. Depending on the type and location of the infection caused by the specific bacterial strain, antisense oligonucleotides can be administered differently by the enteral and parenteral routes. To ensure the ASO chemical compound distribution to a target sequence of the bacterial genome, we designed the attachment of a cell-permeable component for it. It successfully penetrates both eukaryotic and prokaryotic cells, transferring the compound with which it has formed a complex, i.e., the antisense oligonucleotide created by us with a specific target in a bacterial cell pathogen in humans.

## 4. Materials and Methods

### 4.1. Databases Used

To perform the classification of the eight riboswitches, we created a detailed analysis protocol that we applied to each of them. It included various bioinformatic and genomic analyses of the riboswitch’s distribution in the bacterial world, its role in bacterial metabolism, the biochemical pathways it participates in, and its specific characteristics.

We first started with a question about the riboswitches’ spread among the organisms. We used the Rfam 14.8 database (https://rfam.xfam.org/, accessed on 19 June 2022) and Gene Bank from the National Center for Biotechnology Information—NCBI (https://www.ncbi.nlm.nih.gov/, accessed on 19 June 2022). KEGG is a database resource for understanding the high-level functions and utilities of the biological system, such as the cell, organism, and ecosystem, from molecular-level information, especially large-scale molecular datasets generated by genome sequencing and other high-throughput experimental technologies.

To track the distribution of each riboswitch selected for analysis and compare it with the other riboswitches, we selected 60 different bacterial species that have a pronounced negative effect on humans, as mentioned in List 1. It includes some of the most dangerous human pathogens that the World Health Organization has identified as priorities, such as *Acinetobacter baumannii*, *Klebsiella pneumoniae*, *Pseudomonas aeruginosa*, *Salmonella enterica*, *Salmonella typhi*, *Shigella sonnei*, and *Yersinia pestis*. The nucleotide sequences of the riboswitches in individual human pathogenic bacteria were adopted from the Rfam 14.0 database (https://rfam.xfam.org/, accessed on 1 August 2022). All sequences have been sorted and saved in different Fasta files (Appendix A).

### 4.2. Software Used

To detect the local regions of similarity between the sequences from one genus or among all of the presented bacteria for each riboswitch and the human body to the probiotic bacterial flora, we used the ClustalX/ClustalW programs for the multiple and complete alignments, followed by BLASTN. This step determined whether a riboswitch from one or more bacteria was found in the human or probiotic genome. If a match was present, the riboswitch could not be used as a target in subsequent antibacterial drug development and was automatically classified as non-suitable. This was a fundamental step for the following detailed analyses for designing a novel drug agent (Appendix A). An example of the multiple alignment view (Appendix A) for the BLAST of the human pathogenic bacterial riboswitch with the human genome and the genome of the *Lactobacillus* sp.

To present the individual multiple alignments in a more comprehensible and representative way, the ESPript web tool was used. ESPript 3.0. (*Easy Sequencing in PostScript’*) is a program that renders sequence similarities and secondary structure information from aligned sequences for analysis and publication purposes. (https://espript.ibcp.fr/ESPript/ESPript/, accessed on 11 June 2022) (Appendix A).

We have developed software products, such as RevComOligo, Random Oligo Generator, and Motif Searcher programs, available here at https://penchovsky.atwebpages.com/applications.php (accessed on 15 June 2022), which helped us to expand and interact with the analysis [122]. This software was used for the bioinformatics search of riboswitch motifs in different bacteria.

### 4.3. Evaluation procedure

After finishing the previous analyses, we continued with the bioinformatics check of the GenBank of NCBI (https://www.ncbi.nlm.nih.gov/genbank/, accessed on 19 June 2022), Kyoto Encyclopedia of Genes and Genomes (KEGG—https://www.genome.jp/kegg/, accessed on 20 June 2022), and Protein Data Bank (PDB—https://www.rcsb.org/, accessed on 21 June 2022). We used them to examine the known metabolic pathways that are involved in the different riboswitches to answer the three important questions for the four criteria mentioned earlier, to find out whether the riboswitch controls the biosynthetic pathways and transports proteins of the key metabolites with a vital role, and to check for the presence/absence of alternative biosynthetic pathways. We carried out an even deeper analysis of some processes and used the PDB, where we found different proteins and secondary reactions. The results for each of the riboswitches are presented in the Results section.

For the second part of the subsequent biochemical-oriented analyses we used the bioinformatics tools from ExPASy Bioinformatics Resource Portal (https://www.expasy.org/, accessed on 19 June 2022) and the BioCyc database (https://biocyc.org/, accessed on 23 June 2022).

By consistently applying the bioinformatics and genome analyses using resources and collections, as well as various bioinformatics applications and programs, we precisely managed to answer the four questions asked at the beginning of this paper and successfully divided the analyzed eight riboswitches into four groups based on the four criteria. Some of the illustrative materials for the results of the analyses, which are not possible to show here, have been uploaded to the RSwitch database, which is public and free to access (https://penchovsky.atwebpages.com/applications.php?page=58, accessed on 19 August 2022).

We note that our protocol has already been successfully applied to the analysis of another eight bacterial riboswitches, some of which have already shown excellent results in in vitro testing of bacterial drugs already designed and discovered by us [8,45].

## 5. Conclusions

Bacterial riboswitches are increasingly being mentioned as potential targets for developing antibacterial drugs against some of the most severe human pathogenic bacteria. This is due to their advantages, such as their absence in the human body, control of critical metabolites and transport of those essential for the survival and division of the bacterium, as well as their ability to regulate gene expression in several ways. Due to their diversity and depending on their functionalities, they could be recognized and confirmed as suitable targets for developing drugs against pathogenic bacteria or be rejected based on postulated criteria. Based on our bioinformatics and genomic studies, we classified 16 riboswitches, 8 of which are the subject of this paper. Two of them are classified as the most suitable riboswitches—the PreQ1 and the MoCo RNA riboswitches. The riboswitches that fall into the first category will become a priority for laboratory testing to develop antibacterial drugs. They have excellent potential as targets for the design of drugs (antisense oligonucleotides or other substances), which, as a result of specific conformational changes and regulation of gene expression, will lead to a bacteriostatic or bactericidal effect. Very suitable are c-di-GMP I and c-di-GMP II. They will also be tested for drug ability. They control the main biosynthetic pathways for metabolites important to bacteria but do not control alternative synthetic pathways for the same metabolites. If low levels of metabolites are synthesized as a result of alternative compensatory pathways, they may not be sufficient for the bacteria to function, leading to their death or stopping their division. This could be easily determined in the laboratory. The Mn^2+^ sensor is a suitable riboswitch as a drug target according to the research based on the criteria. On the one hand, this class of riboswitches controls the essential metabolites’ main pathway and is taking part in an alternative pathway for their synthesis.

On the other hand, it controls the transport of essential metabolites. To achieve the absence of synthesized and imported key metabolites, it is necessary to either select identical nucleotide motifs from the Mn^2+^ sensor riboswitches controlling transport, the alternative biochemical pathway, and those controlling the main biochemical pathway or to create two or three separate antisense oligonucleotides to be used in parallel, depending on the levels of the metabolites transported or synthesized alternatively. The desired results are possible but with a few peculiarities. The last three analyzed riboswitches—the glycine, Mn^2+^, and fluoride riboswitches—are classified as not suitable riboswitches for drug targeting.

Thanks to the consistent bioinformatics and genomic analyses, taking into account the obtained results and conclusions, several experiments could be organized with each of the appropriate riboswitches on top of a separate bacterial representative in which they are detected. Thus, it is essential to discover and develop many medicines with a broad spectrum of specific effects, preserving intact beneficial microflora. In addition, this will reduce the use of conventional antibiotics and positively affect the process of antibiotic resistance.

We are working intensively in this direction and the classification of the riboswitches shown in the present work is a significant milestone. For easy public access, we have created a novel database known as RSwitch [123]. The RSwitch is a MySQL database implemented on a PHP-based server, which contains information and annotations of 215 bacterial riboswitches from 16 different classes found in 50 human pathogenic bacteria. It is available at the following web address: https://penchovsky.atwebpages.com/applications.php?page=58 (accessed on 15 August 2022).

## Figures and Tables

**Figure 1 antibiotics-11-01177-f001:**
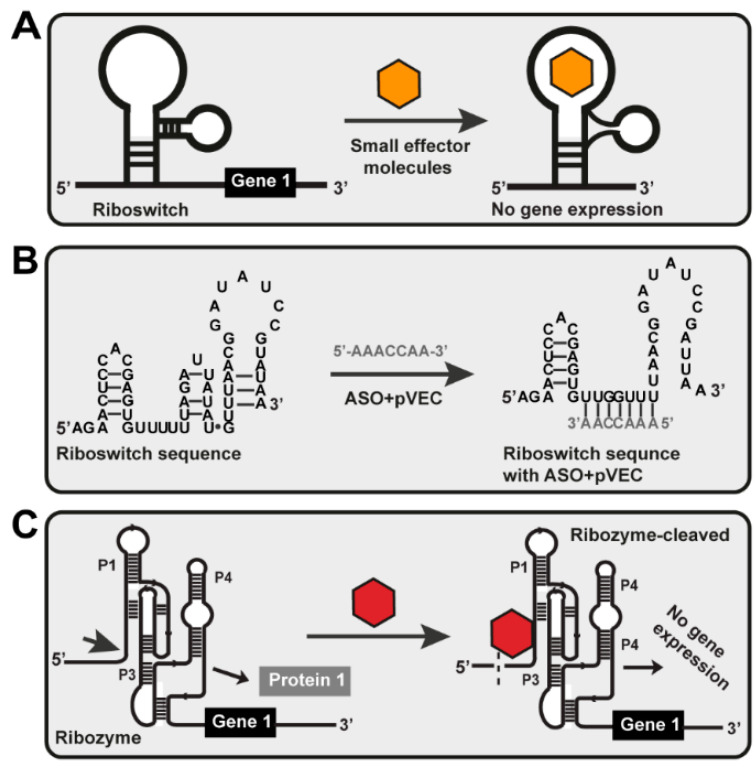
Different riboswitch-targeting mechanisms. (**A**) Small molecule targeting. The effector molecule is a small chemical compound targeted at the riboswitch. When the riboswitch recognizes and binds the small effector molecule to itself, conformational changes occur, disrupting gene expression and blocking protein synthesis. (**B**) ASO targeting. After penetrating the cell with the help of pVEC, ASO is sensed precisely by the riboswitch. It is attached to the targeted sequence and hybridizes entirely complementary to it. That activates a PNase H-dependent hydrolysis, which causes a cleavage in the mRNA structure of the riboswitch. As a result of this cleavage, no gene expression of gene one was observed, i.e., the bacterial cell could not synthesize Protein 1 from gene 1. (**C**) Ribozyme-based ligand binding. Ribozyme is engineered to cleave external RNA molecules under multiple turn-over conditions when sensed by its specific effector molecule. The glmS ribozyme can be made to cleave external RNAs by opening stem P1. It is inactivated without the particular effector molecule for the ribozyme, and the substrate is not cleaved. In the presence of a specific effector, the ribozyme is activated, and the substrate RNA will be cleaved.

**Figure 2 antibiotics-11-01177-f002:**
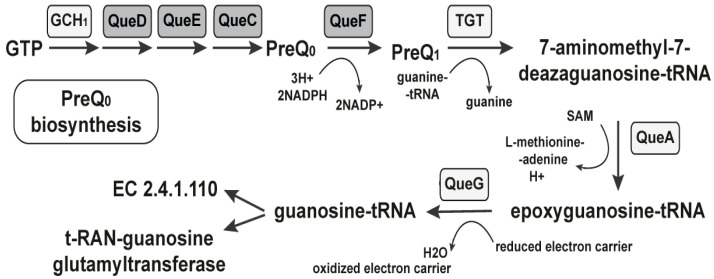
The PreQ1 riboswitch regulates the PreQ1 biosynthesis. Enzymes known to participate in queuosine production are indicated with the required cofactors. Gene products, controlled by the PreQ1 riboswitch, are marked in gray.

**Figure 3 antibiotics-11-01177-f003:**
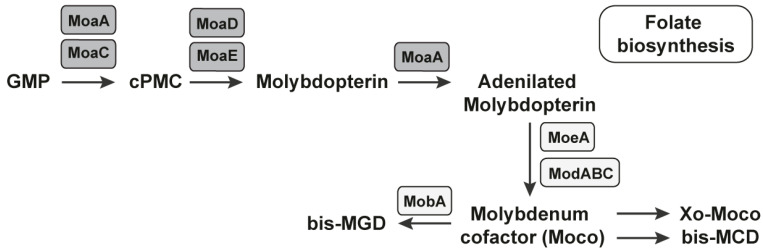
The MoCo RNA riboswitch regulates the biosynthesis of MoCo. All of the mediators/coenzymes under the control of the MoCo RNA riboswitch are presented in the grey rectangles. The biosynthesis of the molybdenum cofactor is realized through four reactions, three of which are under the control of the MoCo RNA riboswitch.

**Figure 4 antibiotics-11-01177-f004:**
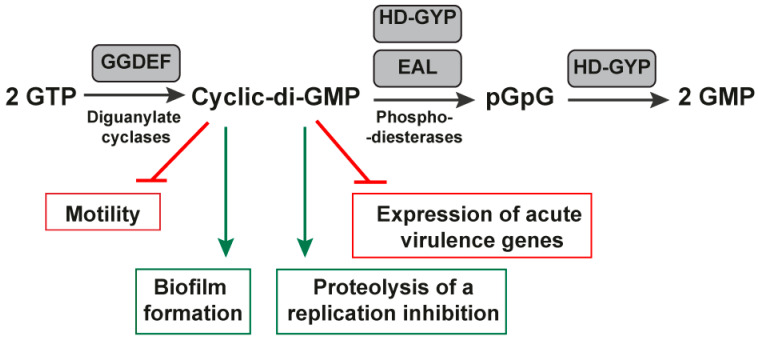
Biosynthesis under the control of cyclic-di-GMP riboswitches. The figure shows the protein domains involved in c-di-GMP metabolism and signaling. The GMP is synthesized by di-guanylate cyclase enzymes (GGDEF). Once synthesized, the c-di-GMP binds to intracellular receptors to decrease motility and increase biofilm formation, contributing to the virulence of several pathogens. Enzymatically active GGDEF, EAL, and HD-GYP domains are in gray rectangles because they control cyclic-di-GMP riboswitches.

**Figure 5 antibiotics-11-01177-f005:**
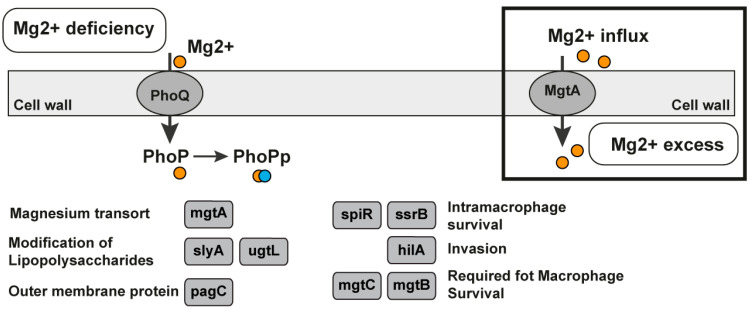
Mg^2+^ riboswitch in *Salmonella enterica*. Magnesium-responsive regulatory pathways in the bacteria *Salmonella enterica* by the two-component regulatory system consisting of PhoQ, the membrane-bound sensor kinase, and PhoP, a transcription regulator. PhoP, which is phosphorylated, affects more than 100 genes in response to fluctuations in the Mg^2+^ concentration. The regulatory system PhoQ/PhoP is a two-component system activated in low magnesium levels or by cationic peptides. Genes regulated by PhoP–PhoQ are depicted in grey boxes and ovals [93].

**Figure 6 antibiotics-11-01177-f006:**
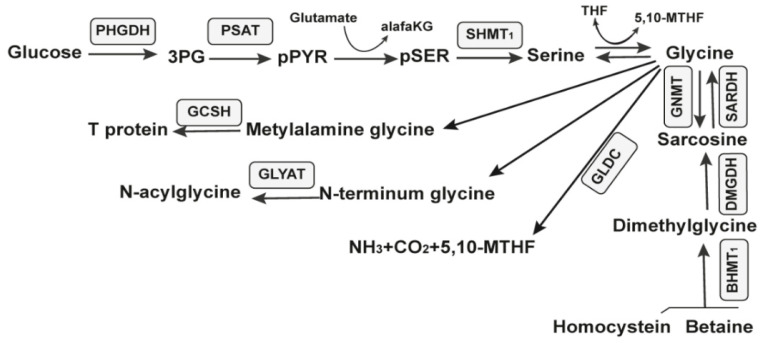
Metabolic pathways of glucose. The ten enzymes involved in the serine, glycine, and sarcosine metabolic pathway are in gray rectangles.

**Figure 7 antibiotics-11-01177-f007:**
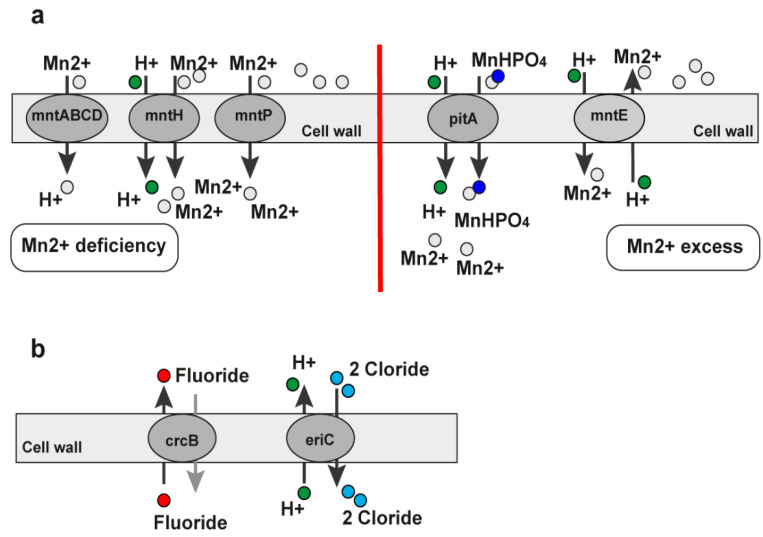
Pathways under the regulation of the Mn^2+^ and fluoride riboswitches. (**a**) MntA pathway regulated by the Mn^2+^ riboswitch. In the presence of Mn^2+^, the *MntR* protein represses the *mntH* transporter. This prevents the transport of Mn and also upregulates the supposed efflux pump. The downstream genes of the mntP promoter are transcribed more than Mn. The addition of the *MntR* protein allows tighter control of the system. In gray ellipses are marked the transporters under the Mn^2+^ riboswitch control. (**b**) Fluoride riboswitch. The fluoride riboswitch regulates genes encoding the proteins crcB and eriC. They are vital for transporting fluoride, hydrogen, and chloride ions.

**Table 1 antibiotics-11-01177-t001:** Cyclic-di-GMP I, Cyclic-di-GMP II, Fluoride, Glycine, Mg^2+^, Mn^2+^, MoCo RNA, and PreQ1 riboswitches are found in human bacterial pathogens.

Human Pathogenic Bacteria		Glycine	PreQ1	Mn^2+^	Fluoride	MoCo RNA	Cyclic-di-GMP I	Mg^2+^ sensor	Cyclic-di-GMP II
**1**	*Acinetobacter baumannii*	+		+	+				
2	*Bacillus anthracis*	+	+	+	+		+		
3	*Bacillus cereus*	+	+	+	+		+		
4	*Bordetella pertussis*	+	+	+					
5	*Brucella abortus*	+							
6	*Clostridium botulinum*	+	+		+		+		+
7	*Clostridium difficile*	+	+		+		+		+
8	*Clostridium perfringens*	+	+		+	+	+		+
9	*Clostridium tetani*	+					+		+
10	*Enterococcus faecalis*		+		+				
11	*Enterococcus faecium*		+		+				
12	*Enterobacter* sp.			+		+		+	
13	*Escherichia coli*			+	+	+		+	
14	*Haemophilus influenzae*	+	+			+			
15	*Klebsiella pneumoniae*				+	+		+	
16	*Listeria monocytogenes*	+	+	+					
17	*Mycobacterium tuberculosis*	+							
18	*Mycobacterium ulcerans*	+							
19	*Neisseria gonorrhoeae*	+	+						
20	*Neisseria meningitidis*	+	+						
21	*Pseudomonas aeruginosa*	+		+	+				
22	*Salmonella enterica*			+		+		+	
23	*Salmonella typhi*					+		+	
24	*Shigella sonnei*					+		+	
25	*Staphylococcus aureus*	+		+					
26	*Staphylococcus epidermidis*	+	+	+					
27	*Staphylococcus saprophyticus*	+	+						
28	*Streptococcus agalactiae*	+	+						
29	*Streptococcus mutans*	+			+				
30	*Streptococcus prenumoniae*	+		+					
31	*Streptococcus pyogenes*	+							
32	*Vibrio cholerae*	+		+		+	+		
33	*Yersinia pestis*			+	+	+			
	**Number**	24	15	14	13	10	7	6	4

Bacterial riboswitches are widely spread in human bacterial pathogens. The eight riboswitches are found in 33 pathogenic bacteria out of 60 known bacterial pathogens infecting humans and leading to severe diseases. The most observed are the glycine and PreQ-1 riboswitches, respectively, in 24 and 15 bacterial pathogens. They are followed by the fluoride riboswitch and the Mn^2+^ riboswitch, found in the genomes of 12 different types of human bacterial pathogens. The rest of the riboswitches are located as follows: MoCo RNA in 10, Cyclic-di-GMP I in 7, Mg^2+^ sensor in 6, and Cyclic-di-GMP II in 4 types of human bacterial pathogens of List 1. The bacteria highlighted in red are in the list of bacteria for which new antibiotics are urgently needed, published in 2017 by the WHO.

**Table 2 antibiotics-11-01177-t002:** Suitability of the cyclic-di-GMP I/II, fluoride, glycine, Mg^2+^, Mn^2+^, MoCo RNA, and PreQ1 riboswitches as antibacterial drug targets.

Riboswitch	Riboswitch-Controlled Biosynthetic Pathway	Transporter Protein for Essential Metabolite	Alternative Biosynthetic Pathway Not under Riboswitch Control	Suitability
**PreQ1**	**✔**	**-**	**-**	**+++**
**MoCo RNA**	**✔**	**-**	**-**	**+++**
**Cyclic-di-GMP I**	**✔**	**-**	**✔**	**++**
**Cyclic-di-GMP II**	**✔**	**-**	**✔**	**++**
**Mg^2+^ sensor**	**✔**	**✔**	**✔/-**	**++/+**
**Glycine**	**-**	**-**	**-**	**-/+**
**Mn^2+^**	**-**	**✔**	**-**	**-/+**
**Fluoride**	**-**	**✔**	**-**	**-/+**

The riboswitches are grouped into four categories: most suitable, very suitable, suitable, and not suitable. Most suitable riboswitches (+++) control essential metabolites without alternative biosynthetic pathways and transport. Very suitable riboswitches (++) control the critical metabolite’s biosynthesis and transport. Suitable riboswitches (+) have alternative biosynthetic pathways and do not control their transport. Not suitable riboswitches (-) are riboswitches that control degradation or do not control the biosynthesis of critical metabolites. The symbol “✔” means the condition is fulfilled in all instances, ✔/- means the condition is fulfilled in some instances only, “-” means the condition is not fulfilled.

## Data Availability

Data is available at: https://penchovsky.atwebpages.com/applications.php?page=58, accessed on 17 August 2022.

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
