# Peer review of "Bioinformatics and Genomic Analyses of the Suitability of Eight Riboswitches for Antibacterial Drug Targets"

_antibiotics, 2022, doi:10.3390/antibiotics11091177_

Round 1
Reviewer 1 Report
This is an interesting research article as there is an enormous need to discover new drugs or vaccines to overcome the problem of the antimicrobial resistance. My comments on the article are the following:
The article is not well organized or arranged, it should be arranged as any research article (i.e Abstract, Introduction, Aim of the work, Material and Methods, Discussion and Conclusion)
The introduction should be clear and short, providing an overview of the research topic, your introduction is too long and seems as a review article.
The authors should summarise and synthesise materials in a way that allows them not only to report, but also to compare, contrast, critically review and comment on what has been said in the literature. In each work, the similarities, differences, strengths and weaknesses need to be identified.
Methods section demonstrates insufficient knowledge of the scientific method, or summarizes the pertinent details in an imprecise or inaccurate manner.
Results section is incomplete or not appropriate for the research questions and methods used
In the discussion section, the authors should provide a more detailed interpretation of the results, confronting other studies in the international scientific literature if feasible. The study's strengths and limitations should be reported.
All abbreviations are to be written in full when they first appear in text.
I would suggest a qualified and experienced English language editor to revise the whole article.
Author Response
Dear Editors of Antibiotics,
We did address all reviews’ comments given in red. In addition, we submit a supplementary section with more results.
We do hope that our manuscript will be accepted for publication in the journal Antibiotics.
We accepted the 100% discount of the journal Antibiotics.
Best regards,
Prof. Robert Penchovsky.
This is an interesting research article as there is an enormous need to discover new drugs or vaccines to overcome the problem of the antimicrobial resistance. My comments on the article are the following:
I do agree with this comment for “there is an enormous need to discover new drugs or vaccines to overcome the problem of the antimicrobial resistance”.
The article is not well organized or arranged, it should be arranged as any research article (i.e Abstract, Introduction, Aim of the work, Material and Methods, Discussion and Conclusion)
I do disagree with this comment on “not well organized or arranged it should be arranged as any research article (i.e Abstract, Introduction, Aim of the work, Material and Methods, Discussion and Conclusion)”. We have just followed the journal’s guidelines
The introduction should be clear and short, providing an overview of the research topic, your introduction is too long and seems as a review article
The introduction is edited to make it clearer and it is shortened. It is two and half pages out of 27. i.e., 9% of the whole paper.
The authors should summarise and synthesise materials in a way that allows them not only to report, but also to compare, contrast, critically review and comment on what has been said in the literature. In each work, the similarities, differences, strengths and weaknesses need to be identified.
DONE
Methods section demonstrates insufficient knowledge of the scientific method, or summarizes the pertinent details in an imprecise or inaccurate manner.
DONE.See the added section in the 2nd reviewer.
Results section is incomplete or not appropriate for the research questions and methods used
In the discussion section, the authors should provide a more detailed interpretation of the results, confronting other studies in the international scientific literature if feasible. The study's strengths and limitations should be reported.
DONE.See the added section in the 2nd reviewer.
All abbreviations are to be written in full when they first appear in text.
DONE: adenosylcobalamin (AdoCbl), flavin mononucleotide (FMN), S-adenosyl methionine (SAM), thiamine pyrophosphate (TPP),
I would suggest a qualified and experienced English language editor to revise the whole article
Done.
Reviewer 2 Report
Reviewer Comments
Antibiotic resistance has led to the prioritization of mining novel therapeutic targets in bacteria. The authors of the manuscript present putative bacterial riboswitches that may be considered as potential drug targets. Bacterial riboswitches are regulatory elements in bacterial mRNA that control gene expression. Given their prevalence across diverse bacterial species, they carry the potential of novel drug targets. The authors identify the suitability of certain riboswitches as drug targets based on the criticality of their metabolic pathways. The authors illustrate the importance of each of the potential riboswitch pathways and cite previous research to corroborate their conclusions. The manuscript is interesting in that it holds the potential of novel drug target discovery in the context of antibiotic resistance. However, the reviewer has several concerns regarding the content of the manuscript, specifically on the Bioinformatics and Genomics analysis performed by the author. The author should address the following comments before the manuscript can be considered for publication.
Major comments
· First, the reviewer is confused by the title of the study. The title “Bioinformatics and Genomic Analyses of the Suitability of Eight Riboswitches for Antibacterial Drug Targets” might lead to the reader to assume that the manuscript includes computational methods that were specifically used to test the suitability of selected riboswitch sequences. However, that is not the case. The authors are attributing the suitability based on the biochemical pathways in which each riboswitch is involved.
· Along the lines of my first comment, the Materials and Methods section should thoroughly describe the Bioinformatics methods (if any) and the corresponding software used for each aspect of the study. Did the authors use any Bioinformatics approach that led them to conclude that the proposed 8 riboswitches are good drug targets? Did the authors study any other riboswitches? What percentage of similarity was found with the human genome of each riboswitch? What parameters were used in BLAST? Which version of BLAST? These details and any others should be included in the Methods section. The results from the Bioinformatics analysis should be included in the Results section. However, if the scope of Bioinformatics analysis involved in this study is limited, then the authors should change their title to something like “A Study on the Suitability of Eight Riboswitches as Antibacterial Drug Targets” or something along those lines.
· Very importantly, did the authors check the extent of sequence similarity of the proposed riboswitch targets to human microbiota? For example, to gut bacteria? If not, then a similar analysis should be performed and included in this study.
· Lines 296-303 (“Thus, structure-based and target-based approaches can be used …”): Where are the details and results of the analysis that the authors performed?
· The authors should include the results corresponding to Lines 305-310 (“We have done multiple alignments and blast analysis…”)
Minor Comments
· In table 2, what do +/- and ✔/- mean? The authors should explain the legend in the table caption.
· Please include reference for Lines 163-165 (“One of the most significant benefits of using bacterial riboswitches …”)
Lines 65-66: Please include the function of the expression platform in a riboswitch.
Author Response
Antibiotic resistance has led to the prioritization of mining novel therapeutic targets in bacteria. The authors of the manuscript present putative bacterial riboswitches that may be considered potential drug targets. Bacterial riboswitches are regulatory elements in bacterial mRNA that control gene expression. Given their prevalence across diverse bacterial species, they carry the potential of novel drug targets. The authors identify the suitability of certain riboswitches as drug targets based on the criticality of their metabolic pathways. The authors illustrate the importance of each of the potential riboswitch pathways and cite previous research to corroborate their conclusions. The manuscript is interesting in that it holds the potential for novel drug target discovery in the context of antibiotic resistance. However, the reviewer has several concerns regarding the content of the manuscript, specifically on the Bioinformatics and Genomics analysis performed by the author. The author should address the following comments before the manuscript can be considered for publication.
Major comments
- First, the reviewer is confused by the title of the study. The title “Bioinformatics and Genomic Analyses of the Suitability of Eight Riboswitches for Antibacterial Drug Targets” might lead the reader to assume that the manuscript includes computational methods that were specifically used to test the suitability of selected riboswitch sequences. However, that is not the case. The authors are attributing the suitability based on the biochemical pathways in which each riboswitch is involved.
Dear reviewer, we - the author collective, defined the title of our publication: as “Bioinformatics and Genomic Analyses of the Suitability of Eight Riboswitches for Antibacterial Drug Targets” because of the bioinformatic methods used during the analysis among all of the known bacterial riboswitches, to select and classify them as suitable or not for antibacterial drug targeting. The present publication is a continuation of our previously published article. We used an analogous analysis methodology with an identical objective but on a different group of bacterial riboswitches. It is entitled: “Genome-wide bioinformatics analysis of FMN, SAM-I, glmS, TPP, lysine, purine, cobalamin, and SAH riboswitches for their applications as allosteric antibacterial drug targets in human pathogenic bacteria”. DOI:10.1080/14728222.2019.1618274, https://pubmed.ncbi.nlm.nih.gov/31079546/.
After carefully reading your entire review, we understand that we need to supplement our article with more details about our applied methods, such as NCBI nucleotide BLAST, Multiple Alignment, etc. All of the information that has been added is marked in the following text.
The text is added:”Fasta files have been created with the sequences from all 15 species and used for detailed multiple alignments with Clustal X (Figures S1 and S2). All collected sequences have been analyzed with the Nucleotide Basic Local Alignment Tool from the National Center for Biotechnology Information (NCBI) (https://blast.ncbi.nlm.nih.gov/Blast.cgi). The analysis showed that the PrerQ1-I riboswitch of each human pathogenic bacteria is not found in the human ge-nome nor the genome of probiotic bacteria from the intestinal tract (Figures S4 and S5). Part of them is available on the RSwitch database (https://penchovsky.atwebpages.com/results_database3.php).”
The text is added:”The multiple alignments and nucleotide blast analyses among sequences of the riboswitch from the different bacteria and the human genome showed that between the bacterial sequences are specific regions that are not found in the human genome, nor probiotic bacteria and could be used as a potential target of the riboswitch-based bacterial drug compound. The results obtained from bioin-formatics and the genomic analysis of its sequences in bacteria can be used to determine one or more specific motifs that will serve as a target for new antibac-terial agents. These agents may be antisense oligonucleotides (ASOs), with a complementary target sequence to the target and a peptide to ensure their pene-tration into the bacterial cell (pVEC or other cell-penetrating protein). After spe-cific binding of the target sequence to the ASO, the gene expression of the gene under the riboswitch's control will be regulated.”
The text is added:”The multiple alignments and BLAST analyses showed that non of the bacterial c-di-GMP are found in the human genome, and the genome of the Lactobacillus sp. Visualization of the results is presented on RSwitch at https://penchovsky.atwebpages.com/results_database3.php. These results confirm the suitability of the riboswitch as a suitable target for new drug development.”
Along the lines of my first comment, the Materials and Methods section should thoroughly describe the Bioinformatics methods (if any) and the corresponding software used for each aspect of the study.
The bioinformatic methods used are mentioned in the Materials and Methods section. They are subsequently described in detail in the same section, and their illustrative figures are added in the supplementary information section.
Did the authors use any Bioinformatics approach that led them to conclude that the proposed 8 riboswitches are good drug targets?
We have used different Bioinformatics approaches like NCBI nucleotide BLAST algorithm and ClustalX/ClustalW for Multiple alignments. Our personally developed software products, such as RevComOligo, Random Oligo Generator, and Motif Searcher (available here: https://penchovsky.atwebpages.com/applications.php), have been used. Detailed information can be found in the updated Materials and Methods section.
The answer to your question is a resounding YES. Bioinformatics analyses are the fundamental first step for classifying the riboswitches as suitable or unsuitable.
All of the collected nucleotide sequences of the different riboswitches in different bacterial organisms must be stored and analyzed to see if some of them have an identical region. After that, this region is checked if it is found in the human genome. If it is present, it could not be used as a potential target for drug discovery, so it will automatically be classified as a not suitable one. If not, the approaches with that riboswitch could be followed by subsequent analyses and tests, which will precisely test another of our criteria for the suitability of the riboswitch as its role for the bacterial metabolism or the transport of essential metabolites. Suppose the NCBI nucleotide BLAST shows a result that there are a lot of similarities between the riboswitch in pathogenic bacteria and probiotic ones. In that case, we have to move into not very suitable, or we have to make more in vivo and in vitro analyses to see the effect of the designed drug against the riboswitch from the pathogenic bacteria.
Did the authors study any other riboswitches?
Previously, we analyzed another 8 riboswitches - MN, SAM-I, glmS, TPP, lysine, purine, cobalamin, and SAH riboswitches. Results were published in 2019 - Genome-wide bioinformatics analysis of FMN, SAM-I, glmS, TPP, lysine, purine, cobalamin, and SAH riboswitches for their applications as allosteric antibacterial drug targets in human pathogenic bacteria”, DOI:10.1080/14728222.2019.1618274,
What percentage of similarity was found with the human genome of each riboswitch? What parameters were used in BLAST? Which version of BLAST?
According to Rfam and NCBI gene bank, there are different organisms with specific sequences found in the different 8 riboswitch classes. For example, PreQ1 is found in the genome of 15 different human pathogenic bacterial types. The mentioned databases have found a huge number of sequences for each of the 15 bacterial types.
The main task of this publication is not to determine the motifs to be used for the precise design and construction of the antisense oligonucleotides with which they are attacked. The research reaches the stage where, from the diversity of riboswitches, we can classify which one would be a better potential target and with which to do all the subsequent analyzes involving the detailed data on the percentage similarity between a particular sequence motif of the bacterial riboswitch and the human genome, or the bacteria from the intestinal microflora.
The motifs from the sequences of the different riboswitches in one bacterium could be very different, so if we talk about the tested examples, we can confirm that there are no similarities between them and the human genome. The percentage is zero, and one of the results have been shown in figure 4 from the supplementary section. Figure 5 from the same section have been shown that there are no similarities between the riboswitch PreQ1 and the human probiotic bacteria Lactobacillus sp. The other results for the rest of the bacteria at the same riboswitch class and the rest of the bacterial classes are equal.
Test is added:”The multiple alignments are available at https://penchovsky.atwebpages.com/results_database2.php. Similar to the results, bacterial riboswitches are not found in the human genome, which means that the first important rule for their possible application is observed.”
These details and any others should be included in the Methods section.
The results from the Bioinformatics analysis should be included in the Results section.
Test is added“To perform the classification of the eight riboswitches, we created a detailed analysis protocol that we applied to each of them. It included various bioinformatic and genomic analyses - of the riboswitch’s distribution in the bacterial world, its role in bacterial metabolism, the biochemical pathways they participate in, and its specific characteristics.
We first started with the question about the riboswitches spread among the organ-isms. We used Rfam 14.8 database (https://rfam.xfam.org/) and Gene Bank from the National Center for Biotechnology Information – NCBI (https://www.ncbi.nlm.nih.gov/).
To track the distribution of each riboswitch selected for analysis and compare it with the other riboswitches, we selected 60 different bacterial species that have a pro-nounced negative effect on humans mentioned in List 1. It includes some of the most dangerous human pathogens that the World Health Organization identifies as a pri-ority, like Acinetobacter baumannii, Klebsiella pneumoniae, Pseudomonas aeruginosa, Sal-monella enterica, Salmonella typhi, Shigella sonnei, and Yersinia pestis. The nucleotide sequences of riboswitches in individual human pathogenic bacteria have been adopted from the Rfam 14.0 database (https://rfam.xfam.org/). All sequences have been sorted and saved in different Fasta files (Figure S1).
To detect local regions of similarity between sequences from one genus or among all of the presented bacteria for each riboswitch and the human body to the probiotic bacterial flora, we have used ClustalX/ClustalW programs for multiple and complete alignments, followed by BLASTN. This step determines whether a riboswitch from one or more bacteria is found in the human or probiotic genome. If a match is present, the riboswitch could not be used as a target in subsequent antibacterial drug devel-opment and is automatically classified as non-suitable. This is a fundamental step for the following detailed analyses to design a novel drug agent (Figure S2). An example of the multiple alignment view (Figures S4 and S5) for the BLAST of human patho-genic bacterial riboswitch with the Human genome and the genome of the Lactoba-cillus sp.
To present the individual multiple alignments in a more comprehensible and repre-sentative way, the ESPript web tool was used. ESPript 3.0. (Easy Sequencing in Post-Script') is a program that renders sequence similarities and secondary structure in-formation from aligned sequences for analysis and publication purposes. (https://espript.ibcp.fr/ESPript/ESPript/). (Figure S3).
We have developed software products, such as RevComOligo, Random Oligo Gener-ator, and Motif Searcher programs, available here at https://penchovsky.atwebpages.com/applications.php, which helped us to expand and interact the analysis [122].
After finishing the previous analyses, we continued with the bioinformatics check of the GenBank of NCBI (https://www.ncbi.nlm.nih.gov/genbank/), Kyoto Encyclopedia of Genes and Genomes (KEGG - https://www.genome.jp/kegg/), and Protein Data Bank (PDB - https://www.rcsb.org/). KEGG is a database re-source for understanding high-level functions and utilities of the biological system, such as the cell, the organism, and the ecosystem, from molecular-level information, especially large-scale molecular datasets generated by genome sequencing and other high-throughput experimental technologies. We used it to check the known metabolic pathways which are involved in the different ri-boswitches to answer the 3 important questions important for the four criteria mentioned before, and does the riboswitch control biosynthetic pathways and transport proteins of key metabolites with a vital role, and the presence/absence of alternative biosynthetic pathways. We did an even deeper analysis of some processes and used the PDB, where we found different proteins and secondary reactions. Re-sults for each of the riboswitches are presented in the Result section.
For the second part of the subsequent biochemical-oriented analyses, we used the bi-oinformatics tools from ExPASy Bioinformatics Resource Portal (https://www.expasy.org/) and the BioCyc database (https://biocyc.org/).
Consistently applying the Bioinformatics and genome analyses, using resources and collections, as well as various bioinformatics applications and programs, we precisely managed to answer the 4 questions asked at the beginning and successfully divide the analyzed 8 riboswitches into 4 groups, based on the 4 criteria. Part of the illustrative materials for the results of the analyses, which are not possible to be shown here, are uploaded on the RSwitch database, which is public and free to use (https://penchovsky.atwebpages.com/applications.php?page=58).
We note that our protocol has already been successfully applied to the analysis of an-other 8 bacterial riboswitches, some of which have already shown excellent results in in-vitro testing of bacterial drugs already designed and discovered by us [8] [123].”
However, if the scope of Bioinformatics analysis involved in this study is limited, then the authors should change their title to something like “A Study on the Suitability of Eight Riboswitches as Antibacterial Drug Targets” or something along those lines.
This is not limited - https://penchovsky.atwebpages.com/applications.php?page=58 also given in the Supplementary section
- Very importantly, did the authors check the extent of sequence similarity of the proposed riboswitch targets to human microbiota? For example, gut bacteria? If not, then a similar analysis should be performed and included in this study.
Dear reviewer, thank you very much for your notification. Results from the NCBI nucleotide BLAST of the bacterial riboswitch and the gut microbiota have been added to the result and the supplementary section. The results show that there is no overlap between the individual species. If we go into detail and look not at the whole bacterial riboswitch but individual parts of it, then minimal matches appear. These can be neutralized by carefully designing a bacterium-specific therapeutic agent that is not complementary to the beneficial bacterium's genome. It is almost impossible to achieve complete dysbiosis, but if this is proven in some rare cases, then supplementation with external cultures of the person would restore good intestinal health.
- Lines 296-303 (“Thus, structure-based and target-based approaches can be used …”): Where are the details and results of the analysis that the authors performed?
DONE.
- The authors should include the results corresponding to Lines 305-310 (“We have done multiple alignments and blast analysis…”)
YES. Multiple alignments are given in the Supplementary section.
Minor Comments
- In table 2, what do +/- and /- mean? The authors should explain the legend in the table caption.
The text is added: “The symbol “✔ “ means the condition is fulfilled in all instances, ✔/- means the condition is fulfilled in some instances only, “-” means the condition is not fulfilled.”
- Please include a reference for Lines 163-165 (“One of the most significant benefits of using bacterial riboswitches …”)
The text is added: “One of the most significant benefits of using bacterial riboswitches is that they are not yet found in the human genome but are widely spread in human bacterial pathogens [23].”
Lines 65-66: Please include the function of the expression platform in a riboswitch.
The text is added: “The expression platform changes its confirmation as a result of ligand binding to the aptamer and change the mRNA expression, unsually by termination of transcription or prevention of translation.”
Reviewer 3 Report
The manuscript on the bacterial riboswitches will interest the scientific community, especially with the aspect of using these riboswitches as future targets for antibiotic resistance.
The manuscript is very comprehensive but the style of writing is hard to follow. Sentences should be kept to a length of a readable minimum and as for the content the manuscript could easily split into to publications.
The figures are valid but show no consistent pattern in display.
Since this manuscript is intended to be a review of riboswitches it should be very easily readable and a good balance between quantity and quality.
Author Response
We added new figures in the suppl. section. The text is added.
Reviewer 4 Report
With the fast development of antibiotic resistance, drug-resistant bacterial infection is becoming an increasing public health problem. This manuscript presents a bioinformatics analysis of bacterial riboswitches to discover novel antibacterial drug targets. The materials and methods are adequately described. The typical bacteria in the manuscript are clinically associated. The results are clearly presented and summarized. Overall, this review provides new insights for developing effective antibacterial targets to treat drug-resistant infections.
Author Response
I do agree with these comments.
Round 2
Reviewer 1 Report
I believe that the revised version of author paper addresses all concerns mentioned in my suggestions in detail. In their replies, the authors have, in my opinion, satisfactorily addressed the issues raised by myself and the other reviewers. I found the article to be interesting and insightful, and particularly well written. The methods, results, and discussion sections were all sufficient and clear. I think the authors in their revised version have tried to put their findings into context.
Author Response
We agree with the comments of the reviewer.
Author Response
Dear Reviewer,
Thank you very much for the following analysis of our work.
- We fully agree with your recommendation to present the results of our study visually as a resource that can be used. Due to the limitations in the present work's volume, we tried presenting a small part of the entire volume of information and absolutely all the resources we created. We are grateful to you for the recommendation of the Dryad resource.
Following your model, we filled in the table with sequences. It is located in the supplementary section and mentioned in Table S1.
|
Sequence ID |
Sequence Length |
Sequence Description |
Nucleotide Sequence |
|
AE017334 |
45 |
PreQ1 riboswitch (Bacillus anthracis) |
AATAACGTGGTTCGAAACCATCCCACGTAAAAAAACTAAGGAGAT |
|
CP001056 |
46 |
PreQ1 riboswitch (Clostridium botulinum) |
AATTGCCTAGTTTTTTATAGAGGGATGGTCTTCGAACCTCTCCCAC
|
|
AM180355 |
44 |
PreQ1 riboswitch (Clostridium difficile) |
TCTGTACTTGTTCGAGAACTCCAAGTATAAAAAACTAAGGAATT
|
|
AE016830 |
43 |
PreQ1 riboswitch (Enterococcus faecalis) |
CTCGGACTGGTTCGGAAACTTCCCAGAATAAAAAACTAAGTATCT
|
|
CP003583 |
28 |
PreQ1 riboswitch (Enterococcus faecium) |
ACTTCCCAGAATAAAAAACTAAGTATCT
|
|
AE004969 |
45 |
PreQ1 riboswitch (Neisseria_gonorrhoeae) |
CGCCCCGTGGTTCGAAAACCTCCCACACTAAAAAACTAAGGAAAC
|
- To comply with your recommendation not to present incomplete graphical figures, we have removed the currently available figure showing the multiple alignment with the Clustal X program. We have left the figure showing the same data in complete form using the ESPript tool.
- We carefully looked at your screenshot from the BLAST program. We set the same parameters - sequence: AATAACGTGGTTCGAAACCATCCCACGTAAAAAAAACTAAGGAGAT, Nucleotide blast, to Lactobacillus (taxid:1578) and again I get a result as follows:
Also, when we did BLAST to check the same riboswitch against Gut Microbiome, with MEGABLAST of Nucleotide BLAST, we saw the same result that there is no significant similarity found:
In addition, consider that when we get to the stage where we're going to design suitable antisense oligonucleotide agents, we're going to take out a stretch of the riboswitch that will be shorter than the whole riboswitch length (about 16-22 nucleotides). The selected region will be again tested for hits in the gut microbiome. Partial similarity does not mean that ASO will have an identical effect on both bacteria.
Hypothetically, suppose we assume that there is a partial match between the sequence from the riboswitch and a section of the genome of a probiotic bacterium. In that case, we should test in practice in bacterial cultures the effect of the designed molecule against the target sequence. If we assume that our chosen motif is found in non-pathogenic bacteria and affects them, too, this will lead to dysbacteriosis. Dysbacteriosis is a process that is reversible by supplementation with pro and prebiotics as well as symbiotics.
- The text in Materials and Methods is tough to follow. The authors should re-write the text by splitting the text into separate sub-sections (each sub-section having a separate title) instead of having all the text together. The sub-sections should be arranged in the order of which the methods/steps were followed. Also, please add a flow diagram illustrating each step in your analysis and what software were used for each step. That should enable the reader to understand your method much better.
DONE. Added: “4.1 Databases used”, “4.2 Software used”, “ 4.3 Evaluation procedure”.
- Also, some text in the Materials and Methods are unnecessary or lack specific context. For example, in lines 823-825 (“We have developed software products, such as RevComOligo, Random Oligo Generator, and Motif Searcher programs, available here at https://penchovsky.atwebpages.com/applications.php, which helped us to expand and interact the analysis [122].”), it is hard to understand when and in what context were these software used. Were they used to mine the riboswitch sequences in the first place? Similarly, in the statement “For the second part of the subsequent biochemical-oriented analyses, we used the bioinformatics tools from ExPASy Bioinformatics Resource Portal (https://www.expasy.org/) and the BioCyc database (https://biocyc.org/).”, what was the specific purpose for using these software? Where are the corresponding results? Why aren’t the results included in the main text? Such statements are ambiguous to interpret without a specific context.
The text is added: “This software was used for bioinformatics search of riboswitch motifs in different bacteria. +
- The authors have not yet included the results from their statement “Thus, structure-based and target-based approaches can be used …”as the reviewer had pointed out earlier.
We edited: Thus, structure-based and target-based approaches can be used to identify the mechanism of synthetic ligands that bind to and regulate complex, folded RNAs [41].

Round 3
Reviewer 2 Report
The authors have addressed all my comments. I do not have any further concerns regarding the manuscript. I recommend the manuscript for publication.